# Depleted Calcium Stores and Increased Calcium Entry in Rod Photoreceptors of the *Cacna2d4* Mouse Model of Cone-Rod Dystrophy RCD4

**DOI:** 10.3390/ijms232113080

**Published:** 2022-10-28

**Authors:** Vittorio Vellani, Giovanna Mauro, Gian Carlo Demontis

**Affiliations:** 1Department of Biomedical, Metabolic and Neural Sciences, University of Modena and Reggio, 41125 Modena, Italy; 2Department of Pharmacy, University of Pisa, 56126 Pisa, Italy

**Keywords:** cone-rod dystrophy, voltage-gated calcium channels, α_2_δ-4 protein, caffeine-sensitive calcium stores, non-selective cationic channels, calcium imaging, patch-clamp

## Abstract

Unidentified pathogenetic mechanisms and genetic and clinical heterogeneity represent critical factors hindering the development of treatments for inherited retinal dystrophies. Frameshift mutations in *Cacna2d4*, which codes for an accessory subunit of voltage-gated calcium channels (VGCC), cause cone-rod dystrophy RCD4 in patients, but the underlying mechanisms remain unknown. To define its pathogenetic mechanisms, we investigated the impact of a *Cacna2d4* frameshift mutation on the electrophysiological profile and calcium handling of mouse rod photoreceptors by patch-clamp recordings and calcium imaging, respectively. In mutant (MUT) rods, the dysregulation of calcium handling extends beyond the reduction in calcium entry through VGCC and surprisingly involves internal calcium stores’ depletion and upregulation of calcium entry via non-selective cationic channels (CSC). The similar dependence of CSC on basal calcium levels in WT and MUT rods suggests that the primary defect in MUT rods lies in defective calcium stores. Calcium stores’ depletion, leading to upregulated calcium and sodium influx via CSC, represents a novel and, so far, unsuspected consequence of the *Cacna2d4* mutation. Blocking CSC may provide a novel strategy to counteract the well-known pathogenetic mechanisms involved in rod demise, such as the reticulum stress response and calcium and sodium overload due to store depletion.

## 1. Introduction

Inherited visual impairments represent a heterogeneous group of rare diseases; the RetNet database (https://sph.uth.edu/RETNET/sum-dis.htm, accessed on 9 June 2022), lists 316 genetic loci involved. Among these, 280 have been assigned to a specific gene, indicating that over 10% of retinal genetic diseases lack a causative gene. Upon restricting the search to non-syndromic degenerative diseases of photoreceptors and pigment epithelial cells, 172 assigned genetic loci correspond to 153 identified genes, i.e., over 10% of responsible genes are presently unknown (reviewed by [1]). At the clinical level, patients with defects in a given locus may receive different clinical diagnoses, ranging from retinitis pigmentosa (RP), Leber congenital amaurosis (LCA), cone or cone-rod dystrophy (CORD), and macular degeneration (MD). Furthermore, due to the heterogeneity of inherited retinal degeneration (IRD), patients receiving the same diagnosis may differ in the time-course and severity of the progressive visual loss [2], eventually leading to severe blindness, with a prevalence of about 10% [3].

On top of genetic heterogeneity and phenotypic variability, the pathogenetic mechanisms linking an identified gene variant with IRD may remain unidentified. Defects in *PDE6B*, coding for the regulatory subunit cGMP phosphodiesterase, a critical enzyme for phototransduction [4], cause an increase in the cGMP content of rod photoreceptors [5,6]. The increased calcium influx [7] caused by the upregulation of cGMP-sensitive channels [8,9] may trigger several apoptotic pathways [10,11,12,13,14], leading to rod demise. However, attempts to protect rods by blocking calcium influx through cGMP-gated channels by the blocking of the voltage-gated calcium channel (VGCC), D-cis-diltiazem, generated controversial findings [15,16,17,18]. It is now accepted that cGMP has pleiotropic effects on rod viability, some generated by the activation of cGMP-gated channels [9,15,19] and others via activation of the cGMP-dependent protein kinase (PKG) [20,21], with cross-talk between PKG activation and the increase in [*Ca^2+^*] (reviewed in [22,23]).

In addition to cGMP-gated channels, VGCC-mediated calcium entry may also contribute to IRD associated with increased cGMP levels [24], but the role of VGCC in IRD remains elusive. In photoreceptors, *CACNA1F* codes for the pore-forming (α) subunit of VGCC mediating calcium influx in the synaptic terminal [25,26], but VGCC also plays a structural role in the assembly of the photoreceptor presynaptic terminal [27]. Defects in *CACNA1F* may cause a non-progressive reduction in night vision in patients, known as congenital stationary night blindness [28,29], suggesting that the lack of functional VGCC does not impair photoreceptor viability. Considering that in nerve cells, Cav1 and Cav2 subfamilies of VGCC [30,31] are thought to assemble as heteromeric complexes between the pore-forming α_1_ and the accessory subunits α_2_δ and β, modulating α_1_ gating and trafficking [32], it is presently unclear how patients [33] and mice [34] bearing a frameshift variant of *Cacna2d4* coding for a truncated version (Tyr802*) of the accessory subunit of VGCC, α_2_δ-4, develop IRD with slow rod-cone degeneration (RCD4, OMIM # 610478, link accessed 17 October, 2022) while the complete loss of *Cacna1f* does not affect rod and cone viability in mice [25]. Furthermore, the lack of α_2_δ-4 has been reported to induce a slight depolarizing shift in VGCC activation and partial suppression of inward current [35] as well as having effects similar to the ones observed in *Cacna1f* mutant mice on synaptic assembly between rods and rod-bipolar cells [35,36]. A missense mutation in α_2_δ-4 coding for an aminoacidic substitution (Arg707His) has also been associated with IRD [37], but another *CACNA2D4* variant coding for a truncated α_2_δ-4 (Arg628X) caused a nonprogressive cone dysfunction [38]. All these observations are consistent with the poor genotype-phenotype correlation in IRD.

There is increasing evidence that α_2_δ proteins may have non-canonical effects, i.e., effects unrelated to VGCC gating/trafficking [32,39,40,41,42] or in cellular compartments distinct from the plasma membrane [43]. The opposite effect on photoreceptor viability of non-sense *Cacnai1f* and *Cacna2d4* may thus result from the pleiotropic effects of α_2_δ-4, beyond its conventional actions on VGCC gating and trafficking.

We have investigated the effects of the non-sense *Cacna2d4* mutation in rod photoreceptors and discovered a significant reduction in the calcium content of internal stores, which upregulates calcium entry via channels distinct from VGCC. These effects reveal an additional and unexpected effect of the α_2_δ-4 mutation on calcium homeostasis, and a novel link with known pathogenetic mechanisms associated with store depletion and increased calcium influx.

## 2. Results

### 2.1. The Cacna2d4 Mutation Reduces Inward Currents through the VGCC of Mouse Rod Photoreceptors

We analyzed the impact of the *Cacna2d4* mutation on rods’ VGCC by voltage-clamp recordings. As shown in Figure 1, small inward currents were evoked by voltage ramps in both WT (Figure 1a) and mutant (MUT) (Figure 1b) rods bathed in saline containing 5 mM Ba^2+^, but in the MUT, the amplitude was reduced and its activation right-shifted compared to the WT rod.

The activation parameter’s maximal conductance (g_Ca_), half-activation voltage (V_0.5_), and inverse slope factor (S) were estimated by the best fits of Equation (1) (see Data Analysis in Section 4) to experimental recordings, and their average values plotted in Figure 1c–e, respectively. In five WT and five MUT rods, mean values (±SEM) were: maximal conductance (g_Ca_) 0.29 (±0.06) and 0.10 (±0.05) nS; half-activation voltages (V_0.5_) −28.17 (±6.16) and −10.52 (±3.20) mV; and inverse slope factors (S) 9.80 (±1.88) and 10.41 (±1.37) mV. Significant differences between WT and MUT were found for V_0.5_ and g_Ca_ (* *p* < 0.05; see Figure legend), indicating that the *Cacna2d4* mutation affects VGCC by shifting its activation to depolarized potentials and reducing its maximal conductance.

### 2.2. Analysis of Cacna2d4 Mutation on Calcium Homeostasis in Non-Synaptic Cell Compartments by Calcium Imaging

The reduced g_Ca_ in MUT rods estimated from patch-clamp recordings may result from impaired VGCC trafficking to the plasma membrane of all rod cell domains. However, the g_Ca_ reduction may also reflect the selective loss of synaptic VGCC, as previously reported in the same animal model for the nearly complete loss of synapse-clustered calcium-dependent chloride channels, Tmem16 [44]. Indeed, upon normalization to membrane capacitance (Cm), which is proportional to the membrane surface area, the difference between average conductance values of WT (14.50 ± 2.80 pS/pF) and MUT rods (6.62 ± 3.29 pS/pF) did not reach statistical significance (t = 1.82014 with 8 df: *p* = 0.10623), suggesting the loss of synaptic channels contributes to the reduction in calcium current amplitudes in MUT rods.

To evaluate the impact of the mutation in selected cell domains, we used the non-ratiometric probe Fluo−4 [45] to measure the calcium changes due to membrane depolarizations obtained by increasing extracellular K ([K]_o_). Figure 2 plots the increase in fluorescence induced in the cell body compartment of WT (Figure 2a) and MUT (Figure 2b) rods in response to membrane depolarization induced by increasing [K]_o_ from 3.6 (Figure 2a″,b″) to 16 mM (Figure 2a‴,b‴).

The time-course of normalized fluorescence changes (F/F_MAX_) in response to [K]_o_ shows a higher increase in the WT (Figure 2c) compared to the MUT (Figure 2d) rod. Responses of the same rods in Figure 2a–d to saline with [K]_o_ ranging from 10 to 40 mM indicate that at any value of [K]_o,_ a higher increase in fluorescence occurs in the WT (Figure 2e) than in the MUT (Figure 2f) rod. Analysis by two-way ANOVA (see Figure legend) of average values from 34 WT (green) and 37 MUT (red) rods plotted in Figure 2g indicates a significant difference between the average responses of WT and MUT rods to 25 (** *p* < 0.01) and 40 mM (*** *p* < 0.001), compared to 10 mM [K]_o_. Calcium imaging data in the cell body domain indicate the difference between WT and MUT rods in g_Ca_ found by patch-clamp recordings could not be fully accounted for by the lack of functional VGCC in the synaptic terminals in MUT mice.

As previously reported in mouse rods [46], calcium imaging also revealed that in both WT and MUT rods, F/F_MAX_ values did not recover to their baseline levels (dashed lines in Figure 2c,d) following the exposure to increased [K]_o_. Moreover, additional increases in F/F_MAX_ above baseline values (dashed lines in Figure 2e,f) did occur in both WT (Figure 2e) and MUT (Figure 2f) rods following the exposure to higher [K]_o_. Although both the WT (Figure 2c,e) and the MUT (Figure 2d,f) rod displayed a buildup of fluorescence during recovery from [K]_o_-induced changes, the increase appears more prominent in the MUT than in the WT. The average F/F_MAX_ values from 34 WT and 37 MUT rods (Figure 2h), measured before the application of saline with increased [K]_o_, indicate baseline values progressively increase with [K]_o_. A two-way ANOVA found significant effects of both genotypes (*p* = 1.08782 × 10^−4^, see Figure legend) and [K]_o_ (*p* = 5.9971 × 10^−4^, see legend). A post-hoc analysis indicates for MUT rods a significant increase in F/F_MAX_ values (*, *p* < 0.05, see legend) preceding the application of 40 mM [K]_o_, i.e., during the recovery from [K]_o_ 25 mM, compared to baseline values preceding the application of [K]_o_ 10 mM.

The more prominent buildup of F/F_MAX_ values in MUT than in WT rods, despite a significantly decreased ΔF/F_MAX_ in response to [K]_o_, suggests that the *Cacna2d4* mutation interferes with [*Ca^2+^*] homeostasis via additional factors distinct from the control over VGCC gating and trafficking, such as the store depletion-induced activation of calcium-permeable channels distinct from VGCC, previously reported in mouse rods [46].

### 2.3. Multiple Mechanisms Contribute to F/F_MAX_ Buildup in WT and MUT Rods

In addition to the calcium influx via VGCC, calcium release from intracellular stores and influx via voltage-independent calcium-permeable cation-selective channels (CSC) could contribute to the buildup of F/F_MAX_ values following the activation of VGCC in WT and MUT rods. To probe the relative contribution of store release or calcium entry via channels distinct from VGCC, at the end of the 60 s-long recovery in 3.6 mM [K]_o_ that follows the exposure to [K]_o_ 25 mM, i.e., during the increase in F/F_MAX_ over baseline level, rods were exposed to saline containing 5 mM of the calcium chelator EGTA with no -added calcium (0 Ca/EGTA saline, see Table 1 in Materials and Methods. In the absence of calcium release from internal stores, the block of calcium influx should bring intracellular calcium to low nanomolar values. Therefore, in order to evaluate whether the internal [*Ca^2+^*] drops down to low nM values, we converted FLUO-4 fluorescence values to [*Ca^2+^*], using Equation (3), by exposing rods to saline containing 10 µM of the calcium ionophore ionomycin and calibrated calcium concentrations as previously reported for light-induced [*Ca^2+^*] changes in mouse rods [47] (see Section 4 for detail).

The double arrow lines labeled 1 in Figure 3 indicate that [*Ca^2+^*] levels failed to recover to baseline levels (dashed traces) during the 60 s-long recovery in 3.6 mM [K]_o_ that follows the response to 25 mM KCl by a WT (Figure 3a) and a MUT (Figure 3b) rod.

The double arrow lines labeled 2 in Figure 3a,b indicate that [*Ca^2+^*] drops below the baseline level in both the WT and MUT rods during a 20 s-long application of the 0 Ca/EGTA saline. However, the decrease attained low nM levels only in the MUT rod. These results may indicate that the increase in [*Ca^2+^*] observed in MUT rods after activation of VGCC results from a prevailing influx of calcium from outside via CSC, while in WT, the release of calcium from internal stores also contributes to the increase in [*Ca^2+^*] above baseline levels. To release calcium from internal stores, at the end of the 20 s-long application of 0 Ca/EGTA saline, rods were exposed for 60 s to 0 Ca/EGTA saline with 10 µM ionomycin added. As shown in Figure 3a, this resulted in an increase in [*Ca^2+^*] (double arrow line labeled 3) in the WT rod, which returned to its values preceding the application of 0 Ca/EGTA saline, while the response appeared strikingly different in the MUT rod (Figure 3b) with a marginal increase in [*Ca^2+^*].

Figure 3c–e plots the relative increase in [*Ca^2+^*] over baseline [*Ca^2+^*] levels ([*Ca^2+^*]_BAS_) (Figure 3a), the relative decrease in [*Ca^2+^*] in 0 Ca/EGTA as a function of [*Ca^2+^*] at the end of recovery ([*Ca^2+^*]_REC_) from 25 mM [K]_o_ (Figure 3d), and the fractional recovery of [*Ca^2+^*] upon releasing calcium from internal stores by ionomycin (Figure 3e). The larger mean relative increase in baseline for eight MUT (red circles) (2.48 ± 0.74) compared to eight WT (green circles) (0.21 ± 0.13) rods in Figure 3c is statistically significant (t = −3.03682 with 14 df: *p* = 0.00888). The difference between mean baseline [*Ca^2+^*] does not differ significantly between WT (477.65 ± 123.66 nM) and MUT (260.43 ± 54.79 nM) rods (t = 1.60603 with 14 df: *p* = 0.13058), consistent with data shown in Figure 2h, indicating baseline F/F_MAX_ values do not differ significantly before activation of calcium influx by increased [K]_o_. In Figure 3d, the mean relative drop in [*Ca^2+^*] in 0 Ca/EGTA from [*Ca^2+^*]_REC_ was significantly larger in MUT (−0.91 ± 0.04) than in WT (−0.52 ± 0.10) rods (t = 3.52768 with 14 df: *p* = 0.00335). In Figure 3e, the difference between average values of the ratio between [*Ca^2+^*] increases in response to ionomycin and the drop in 0 Ca/EGTA in WT (1.20 ± 0.26) and MUT (0.17 ± 0.07) rods were also statistically significant (t = 4.10 with 13 df: *p* = 0.00124), after removing the outlier WT.

Overall, these data suggest that the impact of the *Cacana2d4* mutation on calcium homeostasis extends beyond its classical effects on VGCC, leading to a reduced calcium content of internal stores and an increased calcium influx via calcium-permeable CSC, resulting in a higher [*Ca^2+^*] increase in MUT than in WT rods in response to VGCC activation.

### 2.4. Analysis of Cacna2d4 Impact on Calcium Stores

Considering the role of store depletion-activated channels in setting basal [*Ca^2+^*] levels in mice [46] and amphibian rods [48], we investigated the effects of the mutation on calcium stores, as illustrated in Figure 4.

Figure 4a–d shows the isolated cell bodies of a WT (Figure 4a) and a MUT (Figure 4a′) rod and the fluorescence before (Figure 4b,b′), during (Figure 4c,c′), and after (Figure 4d,d′) the application of 0 Ca/EGTA saline containing 10 µM of the calcium ionophore ionomycin. Fluorescence in the presence of 20 mM Mn^2+^ corresponds to about 100 nM [*Ca^2+^*] [49,50]. These data from isolated cell bodies confirmed the decrease in calcium store content observed in Figure 3 in cell bodies of intact WT and MUT rods, ruling out the possible contribution to fluorescence changes from adjacent inner segment mitochondria [51].

The decrease in internal stores’ calcium concentration may result from reduced stability. In Figure 4f–i, we evaluated store stability by comparing fluorescence changes induced in WT and MUT rods, primed for 90 min in 0 Ca/EGTA either in the absence (Figure 4f,g) or in the presence (Figure 4h,i) of 2 µM thapsigargin, an inhibitor of the calcium ATPase that uptakes calcium from the cytoplasm into the stores [52]. The response to ionomycin is reduced in MUT rods (4g,i) compared to WT (Figure 4f,h), but thapsigargin treatment for 90 min did not cause a complete depletion of calcium stores in either WT or MUT rods. However, upon blocking calcium uptake by thapsigargin, a reduction in calcium store content should be expected in thapsigargin-treated compared to non-treated MUT rods if the mutation is associated with leaky calcium stores. Panels in Figure 4j–m plot, as a function of basal F/F_MAX,_ ΔF/F_MAX_ values measured in response to ionomycin in 96 WT (Figure 4J) and 101 MUT (Figure 4k) rods bathed in 0 Ca/EGTA saline without thapsigargin, or in 29 WT (Figure 4l) and 51 MUT (Figure 4m) rods primed for 90 min with thapsigargin (2 µM) in 0 Ca/EGTA saline. Using one-way ANCOVA to account for ionomycin response dependence from basal F/F_MAX_, the difference between treated and untreated rods was significant for MUT rods (F = 9.96 with 1, 149 df: *p* = 0.001936), while the thapsigargin effect on the response of WT rods was not significant (F = 2.58 with 1, 122 df: *p* = 0.110807).

Considering the dependence of ionomycin response from basal F/F_MAX_ values, we explored the role of calcium-induced calcium release (CICR) on the reduced calcium content of stores by testing the effects of caffeine, which increases the sensitivity of CICR channels to [*Ca^2+^*] [53] and has been reported, in amphibian rods, to mobilize calcium from internal stores of the cell body ([51]) in a [*Ca^2+^*]-sensitive fashion ([54]). Figure 4n–q analyzes the response of calcium stores to caffeine 10 mM, which increased F/F_MAX_ in both a WT (Figure 4n) and MUT rod (Figure 4o), although the fluorescence increase was larger and faster in the WT than in the MUT rod. Circles in Figure 4p plot response amplitudes (ΔF/F_MAX_) recorded from 14 WT (green) and 20 MUT rods (red). The difference between average ΔF/F_MAX_ of WT (0.084 ± 0.017) and MUT (0.019 ± 0.004) rods was highly significant (*** t = 4.40122 with 32 df: *p* = 1.12099 × 10^−4^). Circles in Figure 4q plot ΔF/F_MAX_ as a function of baseline F/F_MAX_, showing an increase in caffeine response amplitude with increasing baseline F/F_MAX_ values in WT (green) rods, as indicated by the best-fitting straight line whose slope is significantly different from 0 (t = 3.53011, with 12 df: *p* = 0.00415), a finding consistent with the response of internal stores to caffeine previously reported in the cell body of salamander photoreceptors ([54]). On the other hand, the response of MUT rods (red) to caffeine did not change with baseline F/F_MAX_ values, with the best-fitting line not significantly different from 0 (t = −0.25071 with 18 df: *p* = 0.80523), suggesting the dysregulation of CICR in MUT rods.

### 2.5. Shaping of the Electrophysiological Profile by Cation-Selective Channels in Mutant Rods

The finding in Figure 3 of increased [*Ca^2+^*] during recovery from 25 mM [K]_o_ in MUT compared to WT rods, despite the reduction in both calcium content in internal stores and calcium influx via VGCC, suggests an upregulation in MUT rods of calcium-permeable CSC distinct from VGCC.

To test this possibility, we compared the electrophysiological properties of WT and MUT rods by patch-clamp recordings. Data in Figure 5a,b plot slowly-activating inward currents (I_h_) in response to 2 s-long voltage steps from −40 to either −90 or −120 mV by a WT (Figure 5a) and a MUT rod (Figure 5b). The difference between current amplitudes at −90 and −120 mV (double arrow line) in Figure 5b indicates a lower membrane resistance in the MUT than in the WT rod.

Circles in Figure 5c plot the average current amplitudes as a function of imposed voltage in seven WT (green) and five MUT rods (red). Average (±SEM) values for R_m_ measured within the range −120/−90 mV were 1.38 ± 0.11 GΩ and 0.95 ± 0.23 GΩ for WT and MUT rods, respectively, and their difference was borderline (t = 1.92225 with 10 df: *p* = 0.0835). We considered that the variability in the larger hyperpolarization-activated current (I_h_) might represent a confounding factor masking the difference in R_m_ between WT and MUT rods contributed by CSC activation. Traces in Figure 5d,e plot currents evoked by the same stimuli as in Figure 5a,b and show a larger ΔI between current amplitudes at −120 and −90 mV in the MUT (Figure 5e) than in WT (Figure 5d) rods bathed in TEA/Ba/Ca saline, designed to block I_h_ and the outward current I_Kx_ [55]. Circles in Figure 5f plot the average current amplitudes recorded in response to membrane voltages ranging from −120 to +40 mV in four WT (green) and three MUT (red) rods. Average R_m_ measured in the range −120/−90 mV were 11.00 ± 1.76 GΩ and 1.99 ± 0.83 in four WT and three MUT rods bathed in TEA-Ba-Cs saline, respectively, a statistically-significant difference (t = 4.1014 with 5 df: *p* = 0.00934). These data indicate that the *Cacna2d4* mutation is associated with the upregulation of a membrane conductance insensitive to outward (TEA) and inward (Ba^2+^, Cs^+^) rectifying current blockers. The observation that the I/V curves of WT and MUT rods cross the voltage axis close to −10 mV suggests that the lower membrane resistance of MUT rods results from the activation of CSC whose reversal potential is close to −10 mV.

To analyze the biophysical basis of CSC, we carried out cell-attached patch-clamp recordings from the cell body of both WT and MUT rods. We recorded single-channel currents in 9 out of 14 WT and 9 out of 13 MUT rods. In a WT rod (Figure 5g) representative of 4 out of 9 patches, small outward current openings occurred at a pipette voltage (Vp) of −50 mV (upper trace), while no channel opening was detected at a Vp of −10 mV (lower trace). In the MUT rod (Figure 5h) representative of 4 out of 9 patches, inward current openings were recorded at a Vp of −10 mV (lower trace), with tiny channel openings at −50 mV (upper trace).

In the WT rod, the analysis of current amplitude distribution at a Vp of −50 mV (Figure 5i) revealed two peaks, corresponding to a closed (0.72 pA) and open (1.09 pA) channel, whose difference gives a single channel amplitude of about 0.37 pA. The lack of channel openings at a Vp of −10 mV (Figure 5j) was consistent with a single peak fitting the current amplitude distribution. In the MUT rod, a single channel amplitude of about 0.5 pA at a Vp of −10 mV results from the difference between three peaks at −0.93, −1.43, and −2.05 pA (Figure 5l), corresponding to closed, one, and two open channels, respectively. At a Vp of −50 mV (Figure 5k), a small shoulder in the current distribution is consistent with small and infrequent channel openings.

In Figure 5m, the linear fits of I/V curves plotting the average single-channel current amplitudes from the cells in Figure 5g,h as a function of transmembrane voltages provide unitary conductance estimates of 5.04 and 13.30 pS for the WT (green) and the MUT (red) rod, respectively. Reversal potentials are close to −50 (WT, near E_Cl_, see Section 4) and +10 mV (MUT). In recordings lasting >5 min from four WT and four MUT rods, average unitary conductance in Figure 5n of 4.08 ± 0.51 and 10.39 ± 1.25 pS for WT are significantly different (**, t = −4.68033 with 8df: *p* = 0.00339). In Figure 5o, the average reversal potentials of −51.67 ± 11.07 (WT—green) and −2.25 ± 5.2 (MUT -red) are significantly different (**, t = −4.02567 with 6 df: *p* = 0.00691). Although we did not specifically measure the ionic selectivity of these channels, the reversal potential close to 0 mV suggests the CSC corresponds to the 10 pS channel, while the current reversing at about −51 mV flows through chloride-permeable channels.

The up-regulation of CSC and the down-regulation of chloride channels at the cell body are consistent with the different electrophysiological profiles of MUT and WT rods. They indicate that the *Cacna2d4* mutation affects multiple ion channels in rods, beyond VGCC activation and trafficking to the synaptic compartment. However, the occurrence in patches from five WT rods of inward currents at Vp of −10 mV, like in MUT, suggests that the remodeling of the electrophysiological profile associated with the mutation affects the chance of activation of channels normally expressed, although silent, also in WT rods.

### 2.6. The Ca^2+^ Buildup in Response to VGCC Activation Correlates with the Increased CSC Activation

Data in Figure 5 suggest that the difference between the electrophysiological profile of WT and MUT rods is a consequence of CSC upregulation in MUT rods. However, the cell-attached recording of CSC in WT rods may indicate that the *Cacna2d4* mutation, rather than inducing CSC expression, causes a functional upregulation of CSC already present in both WT and MUT rods.

Figure 2 and Figure 3 show that CSC upregulation occurs in response to increased [*Ca^2+^*] in MUT cells with depleted calcium stores. Therefore, it should be expected that some WT rods with high baseline [*Ca^2+^*], i.e., having reduced calcium content in their stores according to data in Figure 4, would display a steady increase in fluorescence akin to that observed in the MUT rod in Figure 2f.

In agreement with this notion, traces in Figure 6a,b plot the progressive increase in fluorescence in both a WT (Figure 6a—green) and MUT (Figure 6b—red) rod with initial baseline fluorescence values > 0.4 F/F_MAX_. The rates of fluorescence increases were 6.3 × 10^−5^ and 13.0 × 10^−5^ ΔF/F_MAX_/s in WT and MUT, respectively, as shown by the dashed black straight lines. Circles in Figure 6c plot the rate of fluorescence increase measured in 39 WT (green) and 49 MUT (red) rods before VGCC activation by membrane depolarization induced by the application of 16 mM [K]_o_. The difference between the average rates of 8.90 × 10^−5^ ± 1.58 × 10^−5^ (WT) and 20.68 × 10^−5^ ± 2.29 × 10^−5^ (MUT) (ΔF/F_MAX_)/s were statistically significant (***, t = −4.02532 with 86 df: *p* = 1.22088 × 10^−4^) and consistent with most WT having F/F_MAX_ baseline values < 0.4.

To assess the contribution of CSC to fluorescence as a function of baseline F/F_MAX_, we used MRS-1845 (N-propargyl nitrendipine), an organic blocker of Ca^2+^ entry through CSC [56] that, like SKF 96,365 [57], does not block the VGCC of rods [48]. The application of 15 μM MRS-1845 (black bar above records) partially and reversibly reduced F/F_MAX_ in both a WT (Figure 6d—green) and a MUT (Figure 6e—red) rod, which attained F/F_MAX_ values close to 1. Importantly, the blocker had substantially smaller effects on WT (Figure 6g—green) and MUT (Figure 6h—red) rods that had baseline F/F_MAX_ < 0.35. Circles in Figure 6f,i plot the fluorescence decrease induced in both 9 WT (Figure 6f) and 12 MUT (Figure 6i) rods by MRS-1845 as a function of fluorescence levels before drug application. The dashed straight lines plot the best fit to data with slopes of −0.08 ± 0.02 (Figure 6f) and −0.10 ± 0.01 (Figure 6i) ΔF/F_MAX_ for a unitary increase in baseline F/F_MAX_. To analyze the possible impact on the drug efficacy of differences between baseline F/F_MAX_ values of WT and MUT rods, we used one-way ANCOVA using changes in fluorescence in response to MRS-1845 as a dependent variable and baseline F/F_MAX_ as a concomitant variable (covariate). The analysis indicates non-significant differences between the mean WT (−0.03 ± 0.01) and MUT (−0.04 ± 0.01) fluorescence drop for a unitary change in baseline F/F_MAX_ (F = 0.03 with 1,18 df: *p* = 0.864423) and regression line slopes (F = 0.64 with 1, 17 df: *p* = 0.434748). These results indicate a similar relation in both WT and MUT rods between baseline F/F_MAX_ and the activation of CSC, suggesting that the mutation affects the probability of activation of CSC rather than their number.

## 3. Discussion

The analysis of the impact of calcium regulation at the cell body compartment of mouse rod photoreceptors revealed the effects of the *Cacna2d4* frameshift mutation on calcium stores and calcium permeation pathways, reducing calcium influx via VGCC while upregulating calcium influx via CSC. These results expand the current understanding of α_2_δ channelopathies beyond the selective impairment of α_1_ gating/trafficking to the synaptic compartment. We will now discuss the impact of the mutation on VGCC, calcium stores, and CSC, as well as their significance for rod function and viability.

### 3.1. Effects of the Mutation on VGCC

The *Cacna2d4* mutation causes the loss of α_2_δ-4 in the retinas of *Cacna2d4* mutant mice [44] and tsA-201cells co-expressing α_1F_ with mutated *Cacna2d4* [58]. The loss of α_2_δ-4 may result from the degradation of the truncated *Cacna2d4* transcript by mRNA nonsense-mediated decay, as reported for rod transcripts *BEST1* [59] and *PROM1* [60]. In addition, truncated α_2_δ-4 lacking the δ moiety may fail to anchor to the plasma membrane, leading to its release in the extracellular space, as previously reported for other truncated α_2_δ isoforms [61].

At the functional level, the loss of α_2_δ-4 was associated with the reduction in the maximal conductance of VGCC (g_Ca_) in both MUT mice in Figure 1 as well in tsA-201 cells [58], a finding consistent with the known canonical effects of intact α_2_δ proteins on VGCC trafficking and gating [62]. Patch-clamp measurements do not resolve VGCC distribution across cellular subdomains, and the decrease in the total current may include the loss of synaptic VGCC, as previously reported for calcium-dependent chloride channels, Tmem16a [44]. However, calcium-imaging data in Figure 2, which show a reduction in the fluorescence increase measured at the cell body of MUT rods in response to KCl, suggest that VGCC trafficking to the plasma membrane is also impaired outside the synaptic compartment. Although we may not exclude a reduction in the VGCC unitary conductance in rods, the effects of the *Cacna2d4* mutation on VGCC are similar to those previously reported for the α_2_δ-2 channelopathy [63] in Purkinje cells, where the lack of α_2_δ-2 reduces g_Ca_ without affecting unitary conductance [63]. The mutation also associates with a rightward shift in activation in MUT rods, but not in tsA-201 cells [58]. This possibly reflects the lack of the CaBP4 subunit [64] in heterologously-expressed VGCC, whose half-activation voltage is already right-shifted compared to native VGCC [65,66] and may indicate a functional interaction between CaBP4 and α_2_δ-4. This notion agrees with the rightward shift in VGCC activation observed in the rods of *Cacna2d4* KO mice [35].

The feedback between horizontal cells and rods [65] is known to affect VGCC, but the use of HEPES-buffered medium and light adaptation during recordings are expected to suppress the feedback contributing to the difference between WT and MUT rods.

Overall, the effects on VGCC activation and trafficking appear superimposable in MUT and *Cacana2d4* KO mice, suggesting that the mutation leads to a substantial loss of functional α_2_δ-4 subunits.

### 3.2. Effects of the Mutation on the Electrophysiological Profile of Rods

The remodeling of the electrophysiological profile of rods with the upregulation of CS is a novel and unexpected feature of the α_2_δ-4 channelopathy. We will now discuss the possible identity of the CSC involved in the remodeling.

Mouse rods express members of the *Trp* superfamily, such as *Trpc1-7* [46,67], and there is evidence for *Trpc1* and *Trpc3* translation in amphibian rods [48]. In mouse rods, Trpc1 channels have been proposed to contribute to calcium entry pathways activated by store depletion [46]. In this study, the CSC long openings in MUT rods are reminiscent of Trpc1A channels [68] and the CSC unitary conductance (close to 10 pS) falls in between the 16 pS reported for Trpc1A and the 5 pS of the Trpc1/Trpc5 heteromeric complex [69] and is quite different from other members of the Trpc subfamily (reviewed in [70]). Data from *Trpc1* KO mice [46] have challenged the role of the Trpc1 channel in setting the resting [*Ca^2+^*] at the cell body compartment of WT mouse rods. However, the cell-attached recordings in Figure 5 showing occasional openings also in WT mice suggest that most CSC are silent in WT but become upregulated in MUT rods in response to the calcium increase in cells with depleted calcium stores, as shown in Figure 2 and Figure 3. Therefore, Trpc1 channels such as CSC may not play a role in setting resting [*Ca^2+^*] in WT rods when their calcium stores are full.

CSC are unlikely to represent cation permeation through either VGCC or cGMP-gated channels. First, CSC are upregulated in MUT rods, while VGCC are reduced. Second, the unitary conductance of mouse cGMP-gated channels in the presence of divalent cations [71] is much lower than the 10 pS found for CSC. Although the identities of CSC and chloride channels involved in the effects of the *Cacna2d4* mutation remain to be determined, our results indicate that the CSC properties match those of either homomeric Trpc1A or Trpc1/Trpc5 heteromeric channels rather than the ones of other channels expressed in mammalian rods.

### 3.3. Effects of the Mutation on Ca^2+^ Handling

Calcium imaging using Fluo-4 revealed that nearly all MUT rods in Figure 4 had depleted calcium stores, as shown by the reduced responses to either caffeine or ionomycin, and store depletion may be the critical event upstream of their electrophysiological profile remodeling and dysregulated calcium handling.

In MUT rods, the calcium store content differs significantly in response to priming for 90′ with thapsigargin, a selective endoplasmic reticulum calcium pump inhibitor. This finding indicates the reduced stability of calcium stores in MUT compared to WT rods, whose response to ionomycin was not affected by thapsigargin treatment. A possible mechanism reducing calcium store stability could be an increased sensitivity to [*Ca^2+^*] of CICR channels. However, MUT rods’ response to caffeine revealed that they lack the sensitivity to basal F/F_MAX_ levels of WT rods. This finding suggests that calcium stores in MUT rods are dysfunctional in terms of calcium content and altered sensitivity of CICR channels to [*Ca^2+^*].

The increased calcium influx from the extracellular side measured in most MUT rods may represent the up-regulation of CSC by depleted stores. In WT and MUT rods, the similar slopes of straight lines fitting the response to MRS-1845 as a function of baseline F/F_MAX_ levels suggest that both WT and MUT rods express CSC, although they are functionally silent in most WT rods. Accordingly, differences between WT and MUT rods may lie in the pathway controlling CSC activation rather than in control over CSC expression, a notion consistent with the occurrence of store depletion-activated calcium entry in mouse rods [46].

Although the notion that the lack of α_2_δ-4 correlates with a decrease in calcium store content and triggers the functional upregulation of CSC in MUT rods is consistent with most of our experimental data, it is unclear whether this occurs via a physical interaction between α_2_δ-4 and Ca^2+^ stores. Furthermore, because the α_2_δ-4 isoform has the shortest C-tail within the α_2_δ family (a single AA residue in the cytoplasm), and in case the C-terminal is cleaved to bind a GPI anchor, as shown for other α_2_δ subunits [72], a direct interaction between α_2_δ-4 and Ca^2+^ stores appears difficult to obtain, even in WT mice.

### 3.4. Functional Relevance of [*Ca^2+^*] Homeostasis at the Cell Body Compartment and the Link between Cacna2d4 Mutation and Rod-Cone Dystrophy

The increase in [*Ca^2+^*] over baseline during the recovery after VGCC activation in MUT mice could not be anticipated from the notion of α_2_δ-4 as an ancillary subunit with canonical actions restricted to VGCC gating and trafficking to the plasma membrane (reviewed in [32]). Although the truncation may prevent α_2_δ-4 membrane insertion, causing its loss from the cells [61], MUT mice should be considered different from the *Cacna2d4* KO. MUT mice express and likely process the *Cacnad2d4* transcript, and it is unclear whether the truncated α_2_δ-4 would interfere with other cellular processes, thus leading to effects unrelated to those played by the full-length α_2_δ-4. The data collected in the MUT mice may prove informative on the pathogenetic mechanisms underlying the development of cone-rod dystrophy.

The first point to discuss is that, in both WT and MUT rods, the recovery from the response to VGCC activation did not revert to the baseline level. Cells strive to achieve a lasting increase in [*Ca^2+^*] after VGCC activation at the cell body, suggesting that this increase has a functional role in rods. It is important to note that in rods the cell nucleus mostly fills the cell body compartment, and gene expression in rods is controlled via the Ca^2+^-dependent adenylyl cyclase Adcy1 [73]. These observations suggest that a lasting elevation in [*Ca^2+^*] may play a role in controlling the transcriptional profile of rods. In most WT rods, the [*Ca^2+^*] increase following VGCC activation involves calcium release from stores, while the upregulation of CSC plays a significant role in MUT. In MUT rods, the activation of CSC in cells with depleted stores may lead to positive feedback causing [*Ca^2+^*] to exceed the range of long-lasting [*Ca^2+^*] elevation in the darkness that is critical for excitation-transcription coupling [74] and for entraining transcription with the light-dark cycle [73]. Intriguingly, in mice homozygous for the *Cacna2d2* ducky(2J) mutation, the dysregulation of gene expression has been proposed to result from the reduction in the somatic Ca^2+^ current [75]. However, no direct evidence linking dysregulated calcium handling at the cell body with transcriptional control is presently available in MUT rods.

A second point that deserves attention is the disruption of the balance between voltage-gated and voltage-independent calcium entry at the cell body level. Patch-clamp recordings indicate currents through CSC reverse close to 0 mV, indicating they have a substantial permeability for monovalent cations in addition to calcium. Therefore, CSC upregulation following the activation of VGCC in store-depleted MUT rods leads to an increase in [*Ca^2+^*] along with a substantial sodium load. Due to their small cell volume, mammalian rods have a fast sodium turnover [76], which is critical for calcium extrusion by the sodium, potassium/calcium exchanger [77] at the outer segment. Furthermore, increasing the sodium load may raise the already high metabolic burden of rod cells [78,79] (reviewed in [80,81]), which may prove critical for rod viability. Reducing ion influx via CSC by MRS-1845 may represent a therapeutic strategy for preventing the consequence of CSC upregulation in MUT rods and its possible adverse effects on their energy balance and viability.

It should be noted that prolonged store depletion causes endoplasmic reticulum stress, which may lead to cell demise, as previously reported for cones of Cnga3 KO mice [82,83,84], suggesting that store depletion may trigger photoreceptor demise independently of store depletion-triggered CSC upregulation in cones. However, even prolonged and substantial store depletion in some systems may not induce stress reticulum-induced cell death [85], indicating the need to assess the relative contribution of store depletion and CSC upregulation to rod demise.

## 4. Materials and Methods

### 4.1. Origin of the Animals

The *Cacna2d4* mutant mouse is an animal model that reproduces the mild cone-rod dystrophy described in patients [33] and was generated in Zurich by breeding WT AJ mice with C57BL/10 expressing the Cacna2d4 mutation in exon 25, which generates an early stop codon in exon 26, leading to a truncated α_2_δ_4_ subunit [34]. Scotopic ERG analysis indicated a similar b-wave reduction in the original C57BL/10 and in mice generated by breeding with A/J mice [34]. In order to generate WT and MUT lines with the mixed BL10/AJ background for patch-clamping and gene expression analysis, heterozygous mice from the same litter were crossed to generate homozygous WT and MUT lines. These lines were tested by genotyping, carried out as previously reported [34]. Note that the mutation was initially described as a frameshift mutation in the exon 25 of the *Cacna2d4* gene (c.2367insC, p.Gly790Arg*fs**14) based on the available sequence of *Cacna2d4* [34]. However, the *Cacna2d4* sequence was later updated, and the frameshift mutation has now been annotated as c.2451insC in exon 25 of *Cacna2d4* [58]. WT and MUT phenotypes were also confirmed by assessing the OPL thickness and photopic and scotopic ERG recordings [44]. A comparison between the ERG responses of dark-adapted mice used in this study [44] (stimulus intensity 83.7 Cs*s/m^2^) and *Cacna2d4* KO mice (stimulus intensity 100 Cd*s/m^2^) [35] indicates similar a-wave amplitudes and strong suppression of the b-wave. The photopic ERG shows a reduced b-wave in both the mutant [44] and KO *Cacna2d4* [35] mice.

### 4.2. Access to Food and Water

Mice were kept in a local facility with water and food ad libitum in a 12-h light/dark cycle with an illumination level below 60 lux.

### 4.3. Euthanasia

Mice 1–2 months old were deeply anesthetized using either intraperitoneal urethane (Sigma Aldrich Srl, Milan, Italy) (0.15 mL/10 g of the weight of a 20% (*w*/*v*) solution in 0.9% NaCl (Sigma Aldrich Srl, Milan, Italy) or halothane (Merck Life Science Srl, Milan, Italy) inhalation, followed by cervical dislocation, and the retina quickly (less than 2 min from anesthesia induction) isolated through a corneal slit as previously reported [86]. No difference was observed between urethane and halothane anesthesia.

### 4.4. Patch-Clamp Recordings

Light-adapted rods were recorded either in whole-mount retinas for macroscopic currents [44], using the blind-patch approach [87], or in 150 μm-thick retinal slices [88] for cell-attached recordings. Rods superfused with Locke’s solution (see Table 1) were identified by their unique electrophysiological profile, i.e., the combination of a hyperpolarization-activated current (*I_h_*), a slowly-inactivating outward current (*I_Kx_*) [55], a Ca^2+^-dependent chloride current (*I_Cl(Ca)_*), and small membrane capacitance (2–5 pF), as previously reported for isolated mouse rods [48,86].

Macroscopic currents through VGCC were recorded using 5 mM BaCl_2_ (Sigma Aldrich Srl, Milan, Italy) as the charge carrier in the presence of 30 mM TEA-Cl (Sigma Aldrich Srl, Milan, Italy) and 3 mM CsCl (Sigma Aldrich Srl, Milan, Italy) (TEA-Ba-Cs saline in Table 1) to reduce voltage- (*I_Kx_* and *I_h_*) and Ca^2+^-dependent currents. Due to the small amplitudes of inward currents through VGCC and the noise added by the leak and capacitive transients subtraction protocol associated with voltage steps (not shown), we opted for a 240 ms-long voltage ramp stimulation protocol applied from a holding voltage of −80 mV to +50 mV. In total, 9–16 sweeps recorded at 3 s-intervals were averaged offline to further reduce noise. Currents were recorded with a HEKA EPC-8 amplifier (Heka Elektronic Gmbh, Lambrecht, Germany), low-pass filtered at 1 kHz with a 3-pole Bessel Filter, and digitized at 33 kHz by an Axodata 1200 (Axon Instruments, USA), using pClamp software version 7.0.1.31 (Axon Instruments Inc., Foster City, CA, USA). *I_h_* was activated in Locke’s solution (see Table 1) by 2 s-long voltage steps from a holding of −40 mV, low-pass filtered at 0.2 kHz, and sampled at 1 kHz. Perforated-patch recordings were carried out using pipettes filled with amphotericin B (270 μg/mL, (Sigma Aldrich Srl, Milan, Italy) dissolved in an intracellular solution (see Table 1). Intra and extracellular solution osmolarity were close to 300 mOsm/L. For cell-attached recordings from rod soma, narrow-bore pipette tips (resistance about 25 MΩ) were prepared from borosilicate glass (Hirschmann 50 μL, Hirschmann Laborgerate Gmbh & Co.,Eberstadt, Germany)). No coating was used to lower stray capacitance. Currents were recorded in the cell-attached configuration using the 50 GΩ feedback resistance, filtered at 0.2 kHz before digitizing at 1 kHz. Voltage stimuli were applied by setting pipette voltages (Vp) at values ranging from −10 to −120 mV.

### 4.5. Ca^2+^ Imaging

Retinas were incubated with papain (0.033% for 12 min at 32 °C; Sigma-Aldrich Srl, Milan, Italy) dissolved in Locke’s solution (see Table 1), followed by mechanical dissociation as previously reported [86]. Dissociated cells were seeded on a glass coverslip, coated with either concanavalin A (0.1%; Sigma-Aldrich Srl, Milan, Italy) or D/L poly-lysine bromide (0.01%) (Sigma-Aldrich Srl, Milan, Italy), that formed the bottom of a well in the center of a 35 mm plastic dish. Fluorescence changes upon VGCC activation were similar when using either concanavalin A or poly-lysine coating. Cells were washed with Ca^2+^-containing PBS and incubated with 10 μM Fluo-4-AM (Invitrogen Italia, Rodano, Italy) for 6′ at 37 °C. Similar results were obtained using 5 µM Fluo-4-AM and 15–25′ incubation at 20 °C. At the end of the incubation, the excess probe was rinsed off with PBS (Sigma Aldrich Srl, Milan, Italy). A dish was mounted on the stage of an upright microscope (Nikon E600, Nikon Microscopy, Moncalieri, Italy), equipped with a high-resolution Andor Ixon camera (Oxford Instruments, Abingdon, UK) and filter for Fluo-4 excitation and emission, and superfused with Locke’s saline. Fluorescence was sampled at 0.5 Hz. Depolarization-activated Ca^2+^ influx was triggered by 30 s-long applications of saline with increasing KCl concentrations (10, 16, 25, and 40 mM), carried out through a manifold with programmable valves (CV Scientific, Modena, Italy). Saline solutions with either 10 or 16 mM KCl were obtained by mixing Locke’s solution with 25 mM KCl saline (see KCl 25 Table 1). A total of 40 mM KCl saline was obtained by adding KCl to the 25 mM KCl solution. Fluorescence values were normalized to those obtained in response to 10 μM ionomycin (Sigma Aldrich Srl, Milan, Italy) in 30 mM CaCl_2_, applied at the end of the experiment. Quenching of Fluo-4 fluorescence with 20 mM Mn^2+^ was also used to generate a reference fluorescence value close to 100 nM free Ca^2+^. The calcium content of internal stores of isolated photoreceptors was measured by the application of 10 μM ionomycin dissolved in a modified Locke’s solution without added Ca^2+^ and with 5 mM EGTA (0 Ca/EGTA, see Table 1) to bring [*Ca^2+^*] below 1 nM. Caffeine (5 mM) (Sigma Aldrich Srl, Milan, Italy) and thapsigargin (2 μM) (Sigma Aldrich Srl, Milan, Italy) were dissolved in Locke’s solution. MRS-1845 (Sigma-Aldrich Srl, Milan, Italy) was dissolved as a 5 mM stock solution in DMSO (Sigma-Aldrich Srl, Milan, Italy) and diluted to 15 μM in Locke’s solution at the time of the recordings.

### 4.6. Data Analysis Perforated-Patch Recordings

Up to 16 sweeps acquired in response to voltage ramps from −80 to +50 mV were averaged offline to reduce noise. Next, the tiny linear component of membrane current in the presence of Ba/TEA/Cs was subtracted by fitting a straight line to data corresponding to membrane potentials ranging from −80 to −70 mV. Leak-subtracted currents were then fitted by Equation (1) using Origin version 8.5.1 Pro (Originlab Corporation, Northampton, MA, USA):(1)$$I\left(V\right)=\frac{g_{Ca}}{1+e^{-\frac{\left(V-V_{Ca}\right)}{S_{Ca}}}}(V-E_{rev})+\frac{g_{Cl}}{1+e^{-\frac{\left(V-V_{Cl}\right)}{S_{Cl}}}}\left(V-E_{Cl}\right)$$
where *g_Ca_*, *V_Ca,_* and *S_Ca_* are the maximal conductance, the half-activation voltage, and the inverse slope factor of the inward current through VGCC, respectively. *E_rev_* and *E_Cl_* were kept fixed at +50 and 0 mV, respectively. E_Cl_ corresponds to the chloride reversal potential in perforated-patch recordings with amphotericin B and 140 mM KCl pseudo-intracellular solution.

Cell-attached recordings. A total of 1.9 s-long recording stretches were binned at 0.02 to 0.05 pA intervals and counts/bin were plotted as a function of bin value and fitted with Equation (2) implemented in Origin version 8.5.1 Pro (Originlab Corporation, Northampton, MA, USA):(2)$$C\left(x\right)=\sum_{i}^{N}\frac{A_{i}}{w_{i}\sqrt{\frac{\pi }{2}}}e^{-2\frac{{\left(x-x_{ci}\right)}^{2}}{w_{i}{}^{2}}}$$
where *C(x)* are the counts for the bin *x*, computed as the sum of *N* Gaussian functions, *A_i_*, is the area under the curve-*i*, and *w_i_* is the spread from the peak *x_ci_*. Data were fitted using N ≤ 3 Gaussian components. The difference between peak values, *x_ci_*, returned the single-channel amplitude for each Vp. For every cell, at least three, 1.9 s-long stretches were analyzed for each Vp to obtain a mean value with its standard error of the mean (SEM). Single channel amplitudes were plotted against transmembrane potential, computed from Vp assuming −60 and −40 mV as the average membrane potential of light-adapted WT and MUT rods, respectively, as assessed by perforated-patch recordings. The best fitting straight line slope provided the single channel conductance, and the *X*-axis intercepted its reversal potential.

### 4.7. Ca^2+^ Imaging Analysis

Rod photoreceptors were identified by their characteristic morphology and analyzed using ImageJ (Image J version 1.43—Mac Biophotonics) to compute fluorescence for a unit area over selected ROIs. Fluorescence was normalized to the peak value in the presence of 10 μM ionomycin plus 30 mM Ca^2+^. Rods with ΔF/F_MAX_ smaller than 0.1 in response to 40 mM KCl were discarded.

Conversion of Fluo-4 fluorescence in [*Ca^2+^*] was carried out by applying the 0 Ca/EGTA solution containing ionomycin (10 µM) to permeabilize cell membranes to calcium, followed by the application of the solution with 496 nM free [*Ca^2+^*] and then by the 30 mM [*Ca^2+^*] saline. The free [*Ca^2+^*] in the presence of EGTA was computed using a free online calculator (https://somapp.ucdmc.ucdavis.edu/pharmacology/bers/maxchelator/CaEGTA-TS.htm accessed on 10–25 August 2022) based on the use of CHELATOR [89].

Fluorescence values were converted into [*Ca^2+^*] using Equation (3)
(3)$$\left[Ca^{2+}\right]=K_{d}\frac{F-F_{0}}{F_{MAX}-F_{0}}$$
where *K_d_* represents the dissociation constant of the reaction between Fluo-4 and [*Ca^2+^*], *F* the measured fluorescence, and *F_0_* and *F_MAX_* represent the fluorescence measured, after exposure to ionomycin, in 0 Ca/EGTA and 30 mM [*Ca^2+^*] saline, respectively. Using the [*Ca^2+^*] of calibrated saline, usually the 494 nM calcium, and solving for *K_d_* as a variable, the average *K_d_* value in 8 WT and 14 MUT rods were 990.25 ± 137.58 and 932.79 ± 203.36 nM, respectively. These values are consistent with those reported for fluo-4 in mouse rods at 22 °C and in the presence of Mg^2+^ [47]. The difference between the *K_d_* values was not significant (t = 0.19788 with 20 df: *p* = 0.84514).

### 4.8. Statistical Analysis

Unless specified, data were presented as the mean ± standard error of the mean (SEM), computed using Origin 8.5.1 Pro. Two-way ANOVA for the effects of genotype and potassium concentrations [K]_o_ used to depolarize the cells and activate VGCC was performed in Origin 8.5.1 Pro. Post-hoc tests using Bonferroni’s correction for multiple comparisons were carried out using a free online calculator from Graphpad (accessed 14–29 August 2022 via link https://www.graphpad.com/quickcalcs/posttest1.cfm). Finally, one-way ANCOVA accounting for the effect of basal calcium levels on the effects of a dependent variable was performed using Vassar stats, an online software freely available at http://vassarstats.net/vsancova.html (accessed on 10–25 August 2022).

## 5. Conclusions

The frameshift *Cacna2d4* mutation had the expected effects on VGCC, with reduced maximal conductance and right-shifted activation. However, dysregulated calcium handling in mutant rods, with depleted calcium stores and upregulated calcium and sodium influx via CSC distinct from VGCC, enlighten possible pathogenetic mechanisms linking faulty α_2_δ-4 to the mild cone-rod dystrophy reported in patients homozygous for the *CACNA2D4* truncating mutation [33] and suggest cation-selective channels may be targeted to prevent a metabolic overload by reducing calcium and sodium entry.

## Figures and Tables

**Figure 1 ijms-23-13080-f001:**
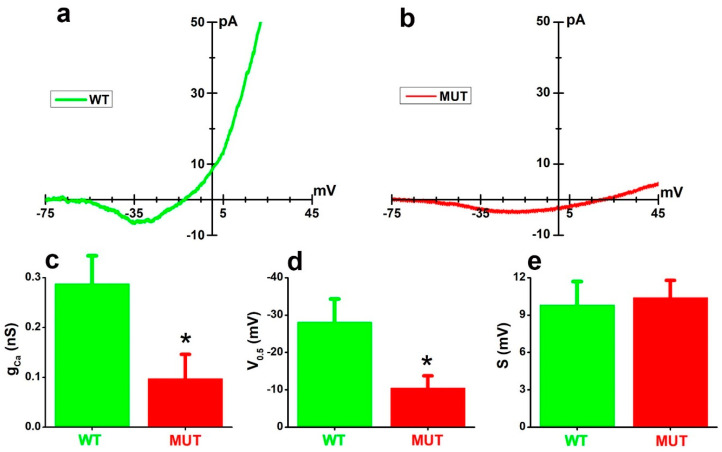
(**a**,**b**) Traces plot average of nine sweeps evoked by voltage ramps in either WT (**a**) or MUT (**b**) rods bathed in TEA-Ba-Cs solution (see Table 1). (**c**–**e**) Bars plot average values (+SEM) for maximal conductance (g_Ca_) (**c**), half-activation voltages (V_0.5_) (**d**), and inverse slope factors (S) (**e**) for five WT (green) and five MUT (red) rods. * *p* < 0.05 by two-tailed *t*-test ((**c**), t = 2.57522, 8 df: *p* = 0.03286) ((**d**), t = 2.54231, 8 df: *p* = 0.03459).

**Figure 2 ijms-23-13080-f002:**
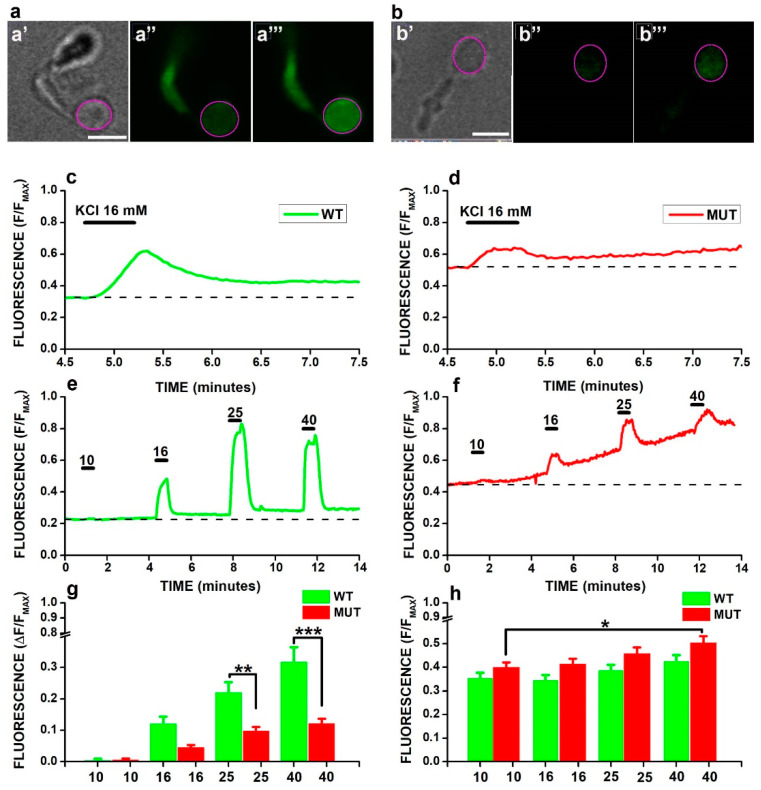
(**a**,**b**) Transmitted light images of isolated WT (**a****′**) and MUT (**b′**) rods and Fluo-4 fluorescence in 3.6 and 16 mM KCl for a WT (**a****″**,**a****‴**) and a MUT (**b****″**,**b****‴**) rod, respectively. Calibration bars in a**′** and b**′** correspond to 5 µm and hold for a and b panels, respectively. Magenta circles in a and b indicate the Region Of Interest (ROI) for fluorescence analysis. (**c**,**d**) Time-course of normalized fluorescence (F/F_MAX_) over the region-of-interest of the cell body (magenta circles in (**a**,**b**)) during a 30 s-long application (thick bars above records) of saline containing 16 mM KCl (black marks above records) for the WT (**c**) and the MUT (**d**) rod. Following exposure to 16 mm KCl, the F/F_MAX_ value of WT and MUT rods did not recover to their baseline levels (dashed traces). (**e**,**f**) F/F_MAX_ changes induced in the WT (**e**) and MUT (**f**) rods above by the application of saline with increasing [K]_o_ concentrations of 10, 16, 25, and 40 mM. Time-course of application is indicated by black tick bars above the records. Note the progressive increase in F/F_MAX_ above baseline (dashed lines) with additional stimulation by increased [K]_o_ in both WT and MUT rods. (**g**) Bars plot mean (+SEM) net increases in normalized fluorescence (ΔF/F_MAX_) induced by saline containing 10, 16, 25, and 40 mM [K]_o_ in 34 WT (green) and 37 MUT (red) rods, respectively. Two-way ANOVA indicated significant effects of both genotype (F = 36.51412, with 1, 275 df: *p* = 4.91296 × 10^−9^) and [K]_o_ (F = 31.05698, with 3, 275 df: *p* = 0). A post-hoc test with Bonferroni’s corrections for multiple comparisons indicated a significant reduction in MUT compared WT rods for [K]_o_ 25 (t = 3.771, MSR = 0.01893; **, *p* < 0.01) and 40 (t = 5992, MSR = 0.01893; ***, *p* < 0.001) mM. (**h**) Bars plot the mean (+SEM) baseline fluorescence (before application of [K]_o_ above 3.6 mM) for WT (green) and MUT (red) rods. The two-way ANOVA indicated significant effects of both genotype (F = 15.42187, with 1, 275 df: *p* = 1.08782 × 10^−4^) and [K]_o_ (F = 5.95736 with 3, 275 df: *p* = 5.9971 × 10^−4^). The post-hoc analysis using Bonferroni’s correction for multiple comparisons indicated a significant baseline increase in MUT rods for baseline values before 40 vs. 10 (t = 3.052, MSR = 0.02036; *, *p* < 0.05) mM [K]_o_.

**Figure 3 ijms-23-13080-f003:**
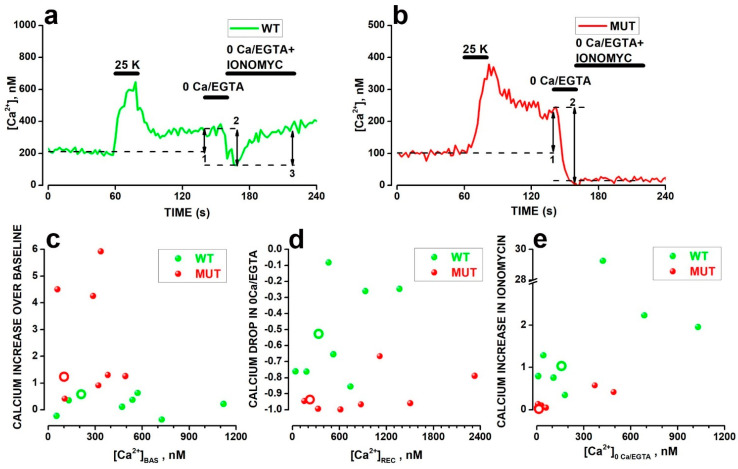
(**a**,**b**) Changes in [*Ca^2+^*] in response to the 20 s-long application of saline with 25 mM [K]_o_ in WT (**a**) and MUT (**b**) rods. Double arrow lines labeled 1: increase in [*Ca^2+^*] after the application of saline with 25 mM [K]_o_. Double arrow lines labeled 2: decrease in [*Ca^2+^*] in response to saline containing no-added calcium and 5 mM EGTA. Double arrow line labeled 3: increase in [*Ca^2+^*] induced by calcium stores’ release in response to 60 s-long application of saline containing no-added calcium, 5 mM EGTA, and the calcium ionophore ionomycin (10 µM) (double arrow line labeled 3 in (**a**)). (**c**–**e**) Circles plot the increase in [*Ca^2+^*] over baseline (**c**), [*Ca^2+^*] drop in 0 Ca/EGTA (**d**), and increase in [*Ca^2+^*] induced by ionomycin in 0 Ca/EGTA (**e**) for eight WT (green) and eight MUT (red) rods. Open circles in (**c**–**e**), plot values measured in rods in (**a**,**b**). The mean relative calcium increases over baseline (([*Ca^2+^*]_RECOV_ − [*Ca^2+^*]_BASELINE_)/[*Ca^2+^*]_BASELINE_)) was significantly different between WT (0.21 ± 0.13) and MUT (2.48 ± 0.74) rods (t = −3.03682, 14 df: *p* = 0.00888). Mean baseline [*Ca^2+^*] levels did not differ significantly between WT (477.65 ± 123.66 nM) and MUT (260.43 ± 54.79 nM) rods (t = 1.60603, 14 df: *p* = 0.13058). The mean relative drop in calcium (([*Ca^2+^*]_0Ca/EGTA_ − [*Ca^2+^*]_REC_)/[*Ca^2+^*]_REC_)) from its level before application ([*Ca^2+^*]_REC_) of 0Ca/EGTA saline was significantly larger in MUT (−0.91 ± 0.04) than in WT (−0.52 ± 0.10) rods (t = 3.52768, 14 df: *p* = 0.00335). The mean ratio between [*Ca^2+^*] increase in response to ionomycin and the drop in 0 Ca/EGTA is significantly larger in WT (1.20 ± 0.26) than in MUT (0.17 ± 0.07) rods (t = 4.10383, 13 df: *p* = 0.00124), after dropping an outlier in the WT group.

**Figure 4 ijms-23-13080-f004:**
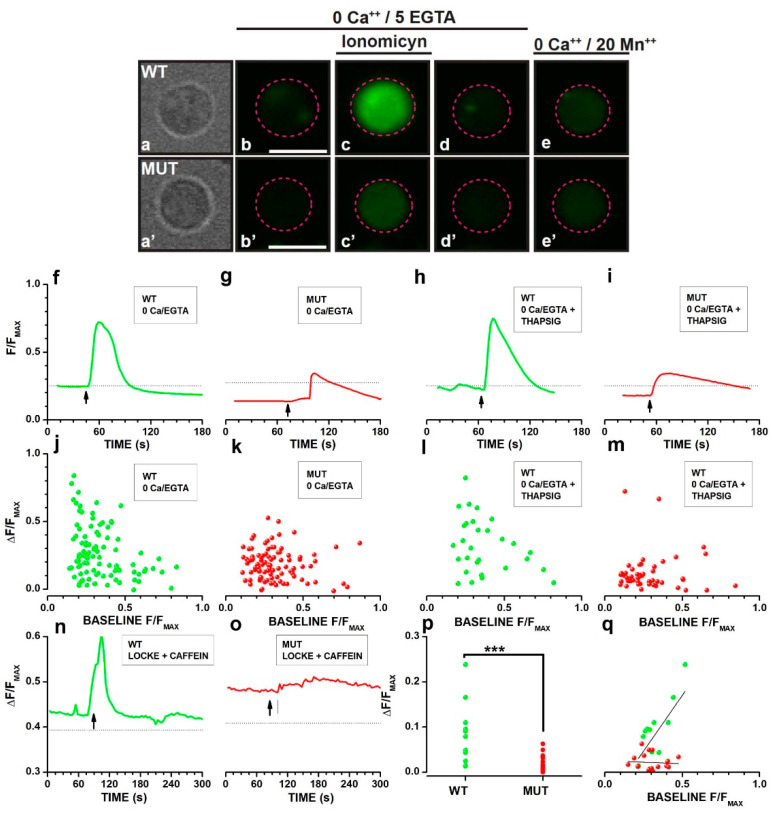
(**a**–**e′**) Bright-field images of isolated cell bodies of a WT (**a**) and a MUT (**a′**) rod. Fluo-4 fluorescence before (**b**,**b′**), during (**c**,**c′**) and after application (**d**,**d′**) of 10 μM ionomycin, and after quenching Fluo-4 fluorescence by 20 mM Mn^2+^ (**e**,**e′**) in a WT (**a**–**e**) and a MUT (**a′**–**e′**) rod. Calibration bars in b and b’ indicate 5 µm and hold for the respective row. Dashed magenta circles plot ROI for fluorescence analysis. (**f**–**i**) Time-course of normalized fluorescence (F/F_MAX_) over the cell body for WT (**f**,**h**) and MUT (**g**,**i**) rods exposed to ionomycin in the absence (**f**,**g**) or after 90 min of priming (**h**,**i**) in Locke’s solution containing 2 µM thapsigargin. Dotted lines plot fluorescence levels in 20 mM Mn^2+^, roughly corresponding to 100 nM Ca^2+^. Upward-pointing arrows indicate the time of the ionomycin puff. (**j**–**m**) Circles plot ionomycin-induced changes in normalized fluorescence (ΔF/F_MAX_) as a function of baseline fluorescence (F/F_MAX_) before ionomycin application for 96 WT (**j**) and 101 MUT (**k**) rods not exposed to thapsigargin or for 29 WT (**l**) and 51 MUT (**m**) rods primed for 90 min in 2 µM thapsigargin. One-way ANCOVA for thapsigargin effect on the response to ionomycin using F/F_MAX_ as a covariate indicates a significant effect of thapsigargin priming on MUT (F = 9.96 with 1, 149 df: *p* = 0.001936) but not WT (F = 2.58 with 1, 122 df: *p* = 0.110807) rods. One-way ANCOVA indicated the slopes of regression lines did not differ significantly between treated and untreated WT (F = 0.03 with 1, 121 df: *p* = 0.862780) or between WT and MUT rods (F = 0.2 with 1, 148 df: *p* = 0.655375). The use of two-way ANCOVA was prevented by regression slopes being significantly different across groups (F = 3.09 with 3, 269 df: *p* = 0.0276), possibly due to the difference between WT and MUT rods. (**n**–**q**) Sweeps plot F/F_MAX_ for a WT (**n**) and a MUT (**o**) in response to 5 mM caffeine (upward pointing arrows). Rods were bathed in Locke’s solution. Dashed lines plot fluorescence levels in 20 mM Mn^2+^. (**p**) Distributions of caffeine-induced responses for 14 WT (green circles) and 20 MUT (red circles) rods. The mean values (±SEM) for WT (0.084 ± 0.017) and MUT (0.019 ± 0.004) were significantly different (t = 4.401, with 32 df: *p* = 1.121 × 10^−4^). (**q**) Circles plot caffeine-induced responses as a function of baseline F/F_MAX_ levels for WT and MUT rods. The dashed lines plot the best fitting linear functions with slopes significantly different from 0 for WT (t = 3.53011 with 12 df: *p* = 0.00415).

**Figure 5 ijms-23-13080-f005:**
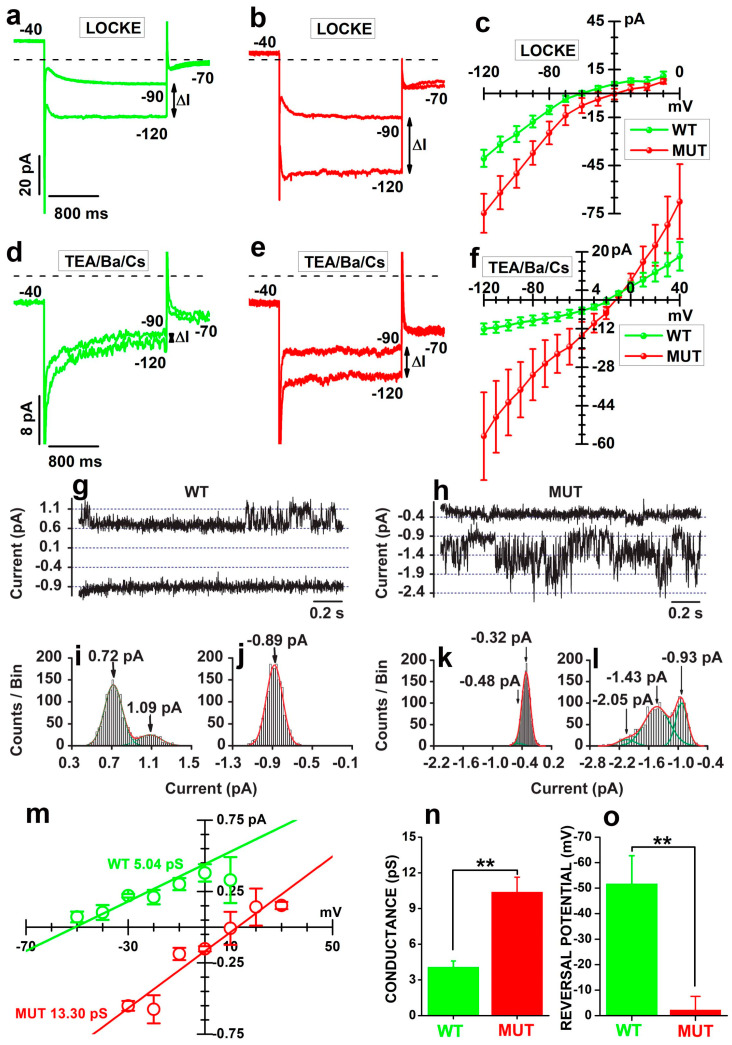
(**a**,**b**) Membrane currents activated in response to 2 s-long voltage steps to −120 and −90 mV in a WT (**a**) and a MUT (**b**) rod bathed in Locke’s solution. In (**a**,**b**,**d**,**e**), numbers close to traces indicate membrane voltage, dashed lines plot the 0-current level, and vertical double arrows lines indicate the increase in current intensities (ΔI) in response to a 30 mV change in membrane potential. (**c**) Circles plot the average current intensities (±SEM) during the last 200 ms of 2 s-long voltage steps, ranging from −10 to −120 mV, as a function of applied voltage (I/V curve) in seven WT (green) and five MUT (red) rods bathed in Locke’s solution. Error bars plot SEM. (**d**,**e**) Membrane currents activated in response to 2 s-long voltage steps to −120 and −90 mV in a WT (**d**) and a MUT (**e**) rod bathed in saline containing TEA/Ba/Cs solution (see Table 1). (**f**) I/V curves for four WT (green) and three MUT (red) rods bathed in a TEA/Ba/Cs solution. Error bars plot SEM. Differences in the mean membrane resistances are computed from the slope of best-fitting straight lines to current amplitudes from −120 to 90 mV of WT (11.00 ± 1.76 GΩ) and MUT (1.99 ± 0.83 GΩ) rods were significant (t = 4.1014 with 5 df: *p* = 0.00934). (**g**,**h**) Currents recorded in cell-attached mode from a WT (**g**) and a MUT (**h**) rod. Pipette voltages (V_p_) of −50 mV (upper sweeps) and −10 mV (lower sweeps). The pipette contained a pseudo-intracellular solution (see INTRA in Table 1). (**i**,**j**) Distributions of current amplitudes (counts/bin) from records in (**g**), for Vp of −50 (**i**) and −10 mV (**j**). Arrows indicate distribution peaks (0.72 and 1.09 pA) in (**i**) and (−0.89 pA) in (**j**). (**k**,**l**) Distribution of current amplitudes (counts/bin) from records in panel (**h**), for Vp of −50 (**k**) and −10 mV (**l**). Arrows indicate distribution peaks (−0.48 and −0.32 pA) in (**k**) and (−0.93, −1.43, and −2.05 pA) in (**l**). In (**i–l**), the red lines plot the best fits of Equation (2) to Counts/Bin data. Bin intervals were 0.02 pA in panels **i**,**j**, and 0.05 pA in (**k**,**l**). (**m**) Circles plot single channel I/V for the WT and MUT rods in (**g**,**h**), respectively. Each point is the average (±SEM) from at least three independent sweeps. Straight lines plot linear regression best fits. (**n**) Bars plot the average single channel conductance (±SEM) of WT (green) and MUT (red) rods. (**o**) Bars plot the average (±SEM) reversal potentials of WT (green) and MUT (red) rods.

**Figure 6 ijms-23-13080-f006:**
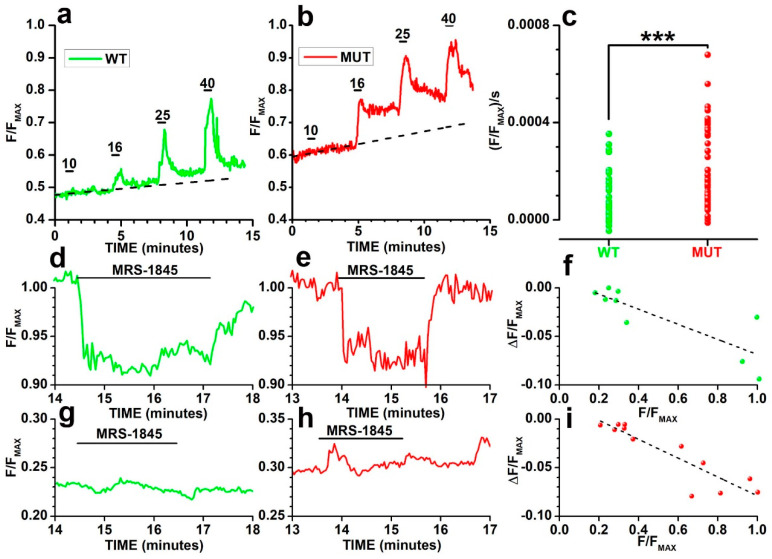
(**a**,**b**) Spontaneous and KCl-evoked changes in fluorescence in a WT (**a**) (green) and in (**b**) MUT (red) rod. VGCC were activated by the 30 s-long application (thick bars above records) of saline with increasing KCl concentrations (10, 16, 25, and 40 mM). Dashed lines plot the best fitting straight lines to sloping baselines. (**c**) Circles plot the distributions of baseline slopes (ΔF/F_MAX_/s/) for 39 WT and 49 MUT rods. A two-tailed t-test indicates a significant difference (t = −4.02532 with 86 df: *p* = 1.22088 × 10^−4^) between mean ΔF/F_MAX_/s of WT (8.90 × 10^−5^ ± 1.58 × 10^−5^) and MUT rods (20.68 × 10^−5^ ± 2.29 × 10^−5^). (**d**,**e**) Changes in normalized fluorescence (F/F_MAX_) in response to the application of 15 μM MRS-1845 (black bars above records) in WT (**d**) (green) and MUT (**e**) (red) rods having high F/F_MAX_ before drug application. (**g**,**h**) Changes in normalized fluorescence (F/F_MAX_) in response to the application of 15 μM MRS-1845 (black bars above records) in WT (**g**) (green) and MUT (**h**) (red) rods having either high F/F_MAX_ before drug application. (**f**,**i**) Circles plot the decreases in normalized fluorescence (ΔF/F_MAX_) induced by 15 μM MRS-1845 as a function of F/F_MAX_ values preceding the application of the blocker in 9 WT (**f**) and 12 MUT (**i**) rods. Filled circles plot the values for the representative rods shown in panels (**f**,**h**). Black lines plot the best fitting linear regression lines. The two-way ANCOVA indicates the mean ΔF/F_MAX_ does not differ significantly between WT and MUT rods (F = 0.03 with 1, 18 df: *p* = 0.864423). The slope of the regression line does not differ significantly (F = 0.64 with 1, 17 df: *p* = 0.434748) between WT and MUT rods.

**Table 1 ijms-23-13080-t001:** The pH was buffered to 7.4 with NaOH 1N for Locke’s and TEA-Ba-Cs solutions and 7.2 with KOH 1N for intracellular saline. Osmolarity was close to 300 mOsm/L for all solutions.

SALINE	NaCl	KCl	CaCl_2_	MgCl_2_	BaCl_2_	TEACl	CsCl	HEPES	PIPES	GLUCOSE	EGTA
LOCKE’S	140	3.6	2.0	2.4	---	---	---	10	----	10	----
TEA-Ba-Cs	100	3.6	0	0	5	30	3	10	----	10	----
INTRA	10	140	0	2.4	----	----	----	----	10	----	----
KCl 25	115	25	2.0	2.4	----	----	----	10	----	10	----
0 Ca/EGTA	140	3.6.	0	0	----	----	----	10	----	10	5
494 nM Ca	140	3.6	4	0	----	----	----	10	----	10	5
30 mM Ca	----	140	30	0	----	----	----	----	10	----	----

## Data Availability

Data presented in this work have been uploaded to a public repository and can be accessed at the link https://doi.org/10.6084/m9.figshare.21206459.v1 (accessed on 21 October 2022).

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
