# Peer review of "Depleted Calcium Stores and Increased Calcium Entry in Rod Photoreceptors of the Cacna2d4 Mouse Model of Cone-Rod Dystrophy RCD4"

_ijms, 2022, doi:10.3390/ijms232113080_

Round 1

Reviewer 1 Report

Comments to authors 

Previous studies reported that mutated α2d4 alters the structure of synaptic terminals in photoreceptors, which then affected the synaptic protein spatial arrangement leading to loss of signal transmission as reported by decreased or loss of ERG b-wave in Cacna2d4 mutant mice with a non-sense mutation. Similarly, deletion of α2δ4 subunits in knockout mice eliminated rod-driven b-waves and reduced cone-driven b-waves, with little or no change in a-waves or photocurrents. Was there any ERG recorded for the mutant mouse model C57BL/10 expressing the frameshift mutation (c.2367insC, p.Gly790Argfs*14) in exon 25 of Cacna2d4? What were the functional differences in that recordings – if ERG recordings were carried out?

The a2d4 subunits are most prominently expressed in rods and cones cells and implicated in presynaptic Cav1.4 regulation based on multiple lines of evidence. In the photoreceptor cells, α2δ4 has been reported to interact with ELFN1 and this interaction is thought to be important for the proper formation of rod synapses. Deletion of α2δ4 or ELFN1 disrupts the formation of rod synapses. In contrast, cone synapses lack ELFN1 and are less strongly affected by the α2δ4 deletion. Have you attempted recording calcium currents in cone cells as they closely share the presynaptic ribbon for downstream phototransduction signaling? Moreover, cone cells are intended for daylight vision which might be less sensitive over the light-adapted protocol.

It has been mentioned that the patch-clamp recordings were recorded on light-adapted rod photoreceptor cells, either in whole retinas or in 150 um-thick retinal slices for macroscopic currents. Rods cells are exclusive for night vision and the cause of light exposure hyperpolarizes the cells as part of its function. When light-adapted rod cells are saturated with light, the rod cells could possibly be less functional to electrical stimulation. Does this experimental condition influence the electrophysiological profile of the rod cells?  - when compared to cone cells.  

Major and Minor comments:

1) Figure 6 is not found and not provided in the manuscript.

2) In Figures 3 a & b, there is a baseline difference in 25 mM KCL evoked [Ca2+] increase on the Y-axis scale. And, for better representation, the use of derivative formulas in the y-axis of figures 3c, 3d & 3e can be replaced with appropriate terminology. The same y-axis formulas are explained redundantly in its legend section as well.

Author Response

Point 1: Previous studies reported that mutated α2d4 alters the structure of synaptic terminals in photoreceptors, which then affects the synaptic protein spatial arrangement leading to loss of signal transmission as reported by decreased or loss of ERG b-wave in Cacna2d4 mutant mice with a non-sense mutation. Similarly, the deletion of α2δ4 subunits in knockout mice eliminated rod-driven b-waves and reduced cone-driven b-waves, with little or no change in a-waves or photocurrents. Was there any ERG recorded for the mutant mouse model C57BL/10 expressing the frameshift mutation (c.2367insC, p.Gly790Argfs*14) in exon 25 of Cacna2d4? What were the functional differences in that recordings – if ERG recordings were carried out?

Response 1: About the scotopic ERG, the comparison between the original mutation in C57BL/10 mice and mice used in this study (generated by breeding the original C57BL/10 with AJ mice) was reported in Wycisk et al., 2006. A comparison between ERG responses of dark-adapted mice used in this study (Caputo et al., 2015 – 83.7 Cs*s/m2) and Cacna2d4 KO mice (Wang et al., 2017 – 100 Cd*s/m2) indicates similar a-wave amplitudes and strong suppression of the b-wave. In addition, the photopic ERG shows a reduced b-wave in the mutant (Caputo et al., 2015) and KO Cacna2d4 (Wang et al., 2017) mice. In the revised version, we added these insights about the previous analysis of the mutant mice in the Material and Methods at the end of section 4.1 Origin of the animals, expanding a previous sentence that reported ERG recording in these mice quoting a previous work.

Q2: The a2d4 subunits are most prominently expressed in rods and cones cells and implicated in presynaptic Cav1.4 regulation based on multiple lines of evidence. In the photoreceptor cells, α2δ4 has been reported to interact with ELFN1, and this interaction is thought to be essential for the proper formation of rod synapses. Deletion of α2δ4 or ELFN1 disrupts the formation of rod synapses. In contrast, cone synapses lack ELFN1 and are less strongly affected by the α2δ4 deletion. Have you attempted recording calcium currents in cone cells as they closely share the presynaptic ribbon for downstream phototransduction signaling? Moreover, cone cells are intended for daylight vision which might be less sensitive over the light-adapted protocol.

A: We agree that the question raised by the reviewer is essential for evaluating the impact of the mutation on synaptic transfer. However, this study focused on the impact of defective Cacna2d4 mutation on calcium homeostasis outside the synaptic compartment and for this reason, we didn’t compare calcium currents in rods and cones. Furthermore, as we mentioned in the introduction, the loss of the pore-forming subunit Cacna1f and the ensuing disorganization of the synaptic terminals didn’t cause a non-progressive loss of night vision, at variance with the progressive visual loss reported in patents with CACNA2D4 mutations. In our opinion, these observations suggest that the loss of synaptic terminals should not represent the most critical event in the progressive loss of photoreceptors and prompted the investigation of the impact of the mutation on calcium homeostasis at the cell body compartment. We would like to remark that the impact of a2d4 variants on calcium homeostasis at the cell body compartment has never been investigated.  

Q3: It has been mentioned that the patch-clamp recordings were recorded on light-adapted rod photoreceptor cells, either in whole retinas or in 150 um-thick retinal slices for macroscopic currents. Rods cells are exclusive for night vision, and the cause of light exposure hyperpolarizes the cells as part of its function. When light-adapted rod cells are saturated with light, the rod cells could possibly be less functional to electrical stimulation. Does this experimental condition influence the electrophysiological profile of the rod cells?  - when compared to cone cells.  

A: Dark-adapted photoreceptors have membrane potentials close to -40 mV, and we kept recorded cells at a holding potential of -40 mV, close to their membrane potential in darkness. Light closes ion channels in the outer segment of photoreceptors, i.e., a region well separated from the cell body. Both calcium and cGMP second messengers are buffered in the outer segment, slowing their diffusion from the OS to the IS/cell body regions. From a technical point of view, light-induced suppression of the dark current reduces background noise in voltage clamp/patch clamp recordings, thus improving the signal-to-noise ratio. We also exploited this principle in Figure 5 by blocking voltage-gated currents to increase membrane resistance and the resolution of remaining currents to reveal the significant difference between the electrophysiological profiles of wt and mut rods.

Major and Minor comments:

Q1) Figure 6 is not found and not provided in the manuscript.

A: We apologize for the lack of figure 6, which has now been included in the revised version.

Q2) In Figures 3 a & b, there is a baseline difference in 25 mM KCL evoked [Ca2+] increase on the Y-axis scale. And, for better representation, the use of derivative formulas in the y-axis of figures 3c, 3d & 3e can be replaced with appropriate terminology. The same y-axis formulas are explained redundantly in its legend section as well.

A: in the revised version, we modified the y-axis label for clarity, using terms defined in the figure legend.

Reviewer 2 Report

The authors investigated the calcium stores depletion and upregulated calcium influx in rod photoreceptors in the cacna2d4 mutation model of mice. Detailed comments are as below:

1. In the methods, how did the authors make sure that there were only the rod photoreceptors being isolated?

2. In figures 2a, 2b and figures 4a-e, 4a’-e’, please add the scale bar in the figure and in the figure legends.

Author Response

The authors investigated the calcium stores depletion and upregulated calcium influx in rod photoreceptors in the cacna2d4 mutation model of mice. Detailed comments are as below:

Point 1. In the methods, how did the authors make sure that there were only the rod photoreceptors being isolated?

Response 1: We treated retinas with papain using conditions that prevented photoreceptors digestion (0.033% for 12 minutes at 32 °C), and retinas were mechanically dissociated to obtain isolated retinal nuerons. We have modified the first sentence in section (4.5 calcium imaging) to avoid conveying the notion of selective isolation of rod photoreceptors. In the first sentence in section 4.7 we added that isolated rods were identified by their characteristic morphology.

Point 2. In figures 2a, 2b and figures 4a-e, 4a’-e’, please add the scale bar in the figure and in the figure legends.

Response 2: Calibration bars have been added in panels 2a, 2b and 4a and 4a’.

Reviewer 3 Report

This manuscript is well written and the experimental design was appropriate to achieve the aim of the study. In my opinion the manuscript do not need any more extra work.

Author Response

We thank you for reviewing the manuscript.